# A Systematic Review of Asthma Phenotypes Derived by Data-Driven Methods

**DOI:** 10.3390/diagnostics11040644

**Published:** 2021-04-02

**Authors:** Francisco Cunha, Rita Amaral, Tiago Jacinto, Bernardo Sousa-Pinto, João A. Fonseca

**Affiliations:** 1Faculty of Medicine, University of Porto, 4200-319 Porto, Portugal; franciscocunha97@gmail.com; 2Center for Health Technology and Services Research (CINTESIS), Faculty of Medicine, University of Porto, 4200-319 Porto, Portugal; tajacinto@gmail.com (T.J.); bernardo@med.up.pt (B.S.-P.); fonseca.ja@gmail.com (J.A.F.); 3Department of Community Medicine, Information and Health Decision Sciences (MEDCIDS), Faculty of Medicine, University of Porto, 4200-319 Porto, Portugal; 4Department of Cardiovascular and Respiratory Sciences, Porto Health School, Polytechnic Institute of Porto, 4200-072 Porto, Portugal; 5Department of Women’s and Children’s Health, Paediatric Research, Uppsala University, 751-05 Uppsala, Sweden; 6Basic and Clinical Immunology Unit, Department of Pathology, Faculty of Medicine, University of Porto, 4200-319 Porto, Portugal; 7Allergy Unit, CUF Porto Hospital and Institute, 4100-180 Porto, Portugal

**Keywords:** asthma, phenotypes, unsupervised analysis, systematic reviews

## Abstract

Classification of asthma phenotypes has a potentially relevant impact on the clinical management of the disease. Methods for statistical classification without a priori assumptions (data-driven approaches) may contribute to developing a better comprehension of trait heterogeneity in disease phenotyping. This study aimed to summarize and characterize asthma phenotypes derived by data-driven methods. We performed a systematic review using three scientific databases, following Preferred Reporting Items for Systematic Reviews and Meta-Analyses (PRISMA) criteria. We included studies reporting adult asthma phenotypes derived by data-driven methods using easily accessible variables in clinical practice. Two independent reviewers assessed studies. The methodological quality of included primary studies was assessed using the ROBINS-I tool. We retrieved 7446 results and included 68 studies of which 65% (*n* = 44) used data from specialized centers and 53% (*n* = 36) evaluated the consistency of phenotypes. The most frequent data-driven method was hierarchical cluster analysis (*n* = 19). Three major asthma-related domains of easily measurable clinical variables used for phenotyping were identified: personal (*n* = 49), functional (*n* = 48) and clinical (*n* = 47). The identified asthma phenotypes varied according to the sample’s characteristics, variables included in the model, and data availability. Overall, the most frequent phenotypes were related to atopy, gender, and severe disease. This review shows a large variability of asthma phenotypes derived from data-driven methods. Further research should include more population-based samples and assess longitudinal consistency of data-driven phenotypes.

## 1. Introduction

Asthma is one of the most common chronic diseases in the world and its prevalence is increasing due to the continuous expansion of western lifestyle and urbanization [1]. Asthma is a chronic inflammatory disease of the airways, characterized by at least partially reversible airway obstruction and bronchial hyper-responsiveness [1,2]. Global Initiative for Asthma (GINA) currently defines asthma as a heterogeneous disease, with a history of respiratory symptoms that vary over time and in intensity, together with variable expiratory airflow [2]. Taking into account that asthma is such a heterogeneous condition with complex pathophysiology, phenotypic classification is essential for the investigation of etiology and treatment tailoring [3]. 

Patients with asthma have been categorized into subgroups using theory- or data-driven approaches. In the classical theory-driven approach, patients with asthma are classified in categories defined a priori according to current knowledge (e.g., based on etiology, severity, and/or triggers) [4]. However, this approach generates asthma phenotypes that are not mutually exclusive, and the correlation with therapeutic response and prognosis might not be the most adequate [5]. 

On the other hand, the data-driven (or unsupervised) approach, which is unbiased by previous classification systems, often starts with a broad hypothesis and uses relevant data to generate a more specific and automatic hypothesis, providing an opportunity to better comprehend the complexity of chronic diseases [4]. Several classes of data-driven algorithms have been involved in tackling the issue of trait heterogeneity in disease phenotyping. The techniques most used to address phenotypic heterogeneity in health care data include distance-based (item-centered, e.g., clustering analysis) and model-based (patient-centered, e.g., latent class analysis) approaches, both of which are not mutually exclusive [6].

Distance-based approaches use the information on the distance between observations in a data set to generate natural groupings of cases [3]. The most commonly used clustering analysis methods are hierarchical, partitioning (k-means or k-medoids), and two-step clustering, which can be roughly described as a combination of the first two. Hierarchical clustering analysis functions by creating a hierarchy of groups that can be represented in a dendrogram, while the partitional methods divide the data into non-overlapping subsets that allow for the classification of each subject to exactly one group [3].

On the other hand, the most used model-based approaches, which use parametric probability distributions to define clusters instead of the distance/similarities between the observations [7], are latent class analysis (LCA), latent profile, and latent transition analysis.

Despite the existence of studies that identified clusters mainly coincident with other larger-scale cluster analyses [8,9,10], there is a lack of consistency of phenotypes and applied methods. Therefore, this systematic review aimed to summarize and characterize asthma phenotypes derived with data-driven methods in adults, using variables easily measurable in a clinical setting. 

## 2. Materials and Methods

In this systematic review, we followed the Preferred Reporting Items for Systematic Reviews and Meta-Analyses (PRISMA) statement [11] and the Patient, Intervention, Comparison and Outcome (PICO) strategy [12] to improve the reporting of this systematic review. 

### 2.1. Search Strategy

Primary studies were identified through electronic database search in PubMed, Scopus, and Web of Science (first search in August 2020; updated in March 2021). Broad medical subject headings (MeSH) and subheadings, or the equivalent, were used and search queries are presented in Table 1. 

### 2.2. Study Selection

Studies were considered eligible when reporting asthma phenotypes determined by data-driven methods in adult patients (≥18 years old), exclusively using variables easily available in a clinical setting. We did not apply exclusion criteria based on language or publication date criteria. Studies using genotyping variables were excluded.

Two authors (F.C. and R.A.) independently screened all the identified studies by title and abstract, after excluding duplicates. Subsequently, potentially eligible studies were retrieved in full-text and assessed independently by two authors, who selected those that met the predefined inclusion and exclusion criteria. Disagreements in the selection process were solved by consensus. Non-English publications were translated if considered eligible. 

Cohen’s kappa coefficient was calculated to evaluate the agreement between the two reviewers in the selection process.

### 2.3. Data Extraction

Two authors (F.C. and R.A.) were involved in data extraction. Study design, setting, inclusion criteria, patients’ characteristics, variables, and data-driven methods used for phenotyping, and the obtained phenotypes, were assessed for each study. 

Variables were divided into eight domains for simplicity and practicality of analysis (Table 2).

### 2.4. Quality Assessment

Two independent researchers (F.C. and R.A.) independently performed the assessment of the quality of the evidence using the ROBINS-I approach [13]. Based on the information reported in each study, the authors judged each domain as low, moderate, serious, or critical risk of bias. Any disagreement was solved by consensus. Quality assessment was summarized in a risk of bias table. 

## 3. Results

### 3.1. Study Selection

A total of 7446 studies were identified in the literature search, of which 2799 were duplicates. After screening all titles and abstracts, which resulted in the exclusion of 4472 records, 175 citations were determined to be potentially eligible for inclusion in our review. Subsequently, full-text assessment resulted in the exclusion of 107 studies in total, including 28 studies incorporating variables or phenotypes with limited applicability in a clinical setting or using phenotypes obtained in previous studies, and 17 studies without available full text. Unavailable references included meeting abstracts, conference papers, posters, and older studies from local publications with no traceable full text. In the end, 68 studies of data-driven asthma phenotypes studies were included. A flowchart for study selection is depicted in Figure 1.

For the selection process, the Cohen’s kappa coefficient and the percentage of the agreement were calculated were determined to be 0.76 and 98%, respectively. These results indicate substantial agreement [14].

### 3.2. Study Characteristics

All the 68 studies [8,9,10,15,16,17,18,19,20,21,22,23,24,25,26,27,28,29,30,31,32,33,34,35,36,37,38,39,40,41,42,43,44,45,46,47,48,49,50,51,52,53,54,55,56,57,58,59,60,61,62,63,64,65,66,67,68,69,70,71,72,73,74,75,76,77,78,79] were published between 2008 and 2020 and recruited patients mostly from specialized centers (*n* = 44, 65%). We identified seven population-based studies. The median sample size of all studies was 249 individuals (range 40–7930).

The included primary studies used a wide variety of methods for cluster analysis, with the most common method being hierarchical cluster analysis (*n* = 19), followed by k-means cluster analysis (*n* = 16) and two-step cluster analysis (*n* = 14). Latent class analysis was the most used model-based approach (*n* = 9) (Figure 2).

It was not possible to retrieve the variables used in two studies [15,16]. The remaining 66 studies of our review were applied a wide range of variables in their respective analysis. Personal variables (e.g., age, gender, BMI, or smoking) were included in the analysis of 74% of the previously mentioned 66 studies. Variables belonging to the lung function, clinical, and atopy domains were all used in more than half of these studies. Figure 3 shows the percentage of studies that used each one of the represented domains of variables.

The characteristics of the 68 studies included in our review are summarized in Table 3. 

### 3.3. Asthma Phenotypes

The number of phenotypes per study ranged from two to eight with a median of four, obtained in 23 studies (34%). A majority of studies (82%) identified between three and five phenotypes. The most frequent phenotypes in our analysis were atopic asthma, severe asthma, and female asthma with multiple variants. 

We observed that 36 studies (53%) evaluated the consistency of phenotypes based on at least one of the following criteria: longitudinal stability, cluster repeatability, reproducibility, and/or validity. 

A visual representation of the variables used for phenotyping by each study is portrayed in Table A1 (Appendix A). Studies with an assessment of consistency are highlighted.

Table 4 represents the defining variables of phenotypes obtained by each study. The full phenotypes are compiled in Table A2 (Appendix A). The results are stratified by a data-driven method, and the frequency of phenotypes in the sample is presented for each study.

In hierarchical cluster analysis, the most frequent phenotypes were atopic/allergic asthma, mentioned 24 times in 13 studies, and late-onset asthma, mentioned 19 times in 12 studies. A common association with atopic asthma was the early age of onset, while late-onset asthma was recurrently linked with severe disease. Atopic asthma was also the most frequent phenotype in two-step cluster analysis. In both k-means and k-medoids cluster analysis, severe asthma occurred the most often. 

In model-based methods, latent class analysis studies identified mostly phenotypes related to symptoms. Factor analysis used severity of disease to classify asthma, while latent transition analysis used allergic status and symptoms. One study derived longitudinal trajectories in terms of pulmonary function using latent mixture modeling.

### 3.4. Risk of Bias Assessment

We used the ROBINS-I tool to assess the risk of bias. The methodological quality of the studies was predominantly moderate (*n* = 29). Of the 68 included studies, 18 were considered to be at overall low risk of bias, while other 18 studies were considered to be at serious risk of bias. Only three studies were judged to be at critical risk of bias. The results are portrayed in Table 5. 

The studies included in our review were in accordance with most of the Strengthening the Reporting of Observational Studies in Epidemiology (STROBE) checklist items [80].

## 4. Discussion

### 4.1. Main Findings

This systematic review revealed a high degree of variability regarding the data-driven methods and variables applied in the models among the studies that identified data-driven asthma phenotypes in adults. There was a lack of consistency in the studies concerning the study setting, target population, choice of statistical method and variables, and ultimately, the label of the phenotype. Overall, the most frequent phenotypes were related to atopy, gender (female), and severe disease.

Different statistical methodologies were applied among the included studies, with hierarchical and k-means clustering being the most common ones. The earliest study in this review (2008) applied a two-step clustering approach to two different sets of patients [33]. In the group of patients of the primary care setting, three phenotypes were determined, namely, “early-onset atopic asthma”, “obese, non-eosinophilic asthma”, and “benign asthma.” In the group of patients with refractory asthma managed in secondary care, four phenotypes were obtained “early onset atopic asthma”, “obese, non-eosinophilic asthma”, “early onset symptomatic asthma with minimal eosinophilic disease”, and “late-onset, eosinophilic asthma with few symptoms” [33]. These phenotypes persisted in later studies, with different variants [8,15,42,55].

Most of the studies recruited patients from specialized centers. However, we identified two population-based studies with a low risk of bias, both using model-based statistical techniques [20,25]. Amaral et al. identified different classes of allergic respiratory diseases using latent class analysis in a population of 728 adults. The study obtained seven phenotypes, which were distinguished according to allergic status and degree of probability of nasal, ocular, and bronchial symptoms [20]. Boudier et al. applied latent transition analysis with nine variables covering personal and phenotypic characteristics on longitudinal data of 3320 adult asthmatics, determining seven phenotypes characterized by the level of asthma symptoms, the allergic status, and pulmonary function. These results revealed strong longitudinal stability [25]. 

There were four population-based studies with some identifiable validation process. Amaral et al. derived phenotypes independently for two age groups and found similar proportions in both age groups for the two obtained data-driven subtypes (“highly symptomatic with poor lung function”, and “less symptomatic with better lung function”), and for previously defined hypothesis-driven subtypes. However, the set of variables was suboptimal to differentiate asthma subgroups [19]. Makikyro et al. applied latent class analysis to identify four asthma subtypes in women and three subtypes in men. Phenotypes were classified according to the control and severity of the disease. The subsequent addition of a set of covariates verified the accuracy of results [50].

An improvement of the characterization of asthma heterogeneity is an essential step in the development of more personalized approaches to asthma management and therapy. There is a need for further research to produce population-based studies with analysis of the longitudinal consistency of data-driven phenotypes. Ilmarinen et al. performed clustering on longitudinal data of Finnish patients with adult-onset asthma. Their approach with 15 variables resulted in the determination of five phenotypes with longitudinal stability, namely “nonrhinitic asthma”, “smoking asthma”, “female asthma”, “obesity-related asthma”, and “early onset atopic adult asthma” [35]. Furthermore, Khusial et al. identified a set of five phenotypes with longitudinal stability in a primary care cohort of adult asthmatics: “smokers”, “late-onset female asthma”, “early atopic asthma”, “reversible asthma” and “exacerbators” [39]. Certain similarities with the results of the study by Ilmarinen et al. are identifiable.

Hsiao et al. found a higher risk of asthma exacerbations in current smoker and ex-smoker clusters in males, as well as in atopy and obesity clusters in females [34]. Park et al. observed an association between smoking males and reduced lung function [57].

The most used dimensions were variables regarding personal, clinical, and functional data. However, other dimensions were used in several studies. For example, Lefaudeux et al. demonstrated that clustering based on clinicophysiologic parameters can produce stable and reproducible clusters [48]. Deccache et al. aimed to characterize treatment adherence with a multidimensional approach encompassing asthma control, attitude towards the disease, and compliance with treatment [29]. Finally, Labor et al. aimed to assess the association of specific asthma phenotypes with mood disorders—five phenotypes were identified by cluster analysis of cross-sectional data in a sample of adult patients of a tertiary center: “allergic asthma”, “aspirin-exacerbated respiratory disease”, “late-onset asthma”, “obesity-associated asthma”, and “infection-associated asthma” [46].

An ongoing investigation is being conducted to identify novel targets and biomarkers for a better understanding of the pathophysiology of asthma. Eventually, the broader availability of emerging molecular and genetic tools may complement the traditional clinical variables in the determination of asthma phenotypes [81].

### 4.2. Strengths and Limitations

We should note that this study has limitations. In an attempt to assemble a complete overview of data-driven asthma phenotyping, some of the included studies focused on specific contexts, which hampered their external validity. Another limitation concerns the possibility of selection bias, as the definition of asthma varied across the studies (questionnaire-based and/or functional-based). This may possibly have implications on selection bias for participant selection and information bias if there are wrong classification and assessment of participants. Other important limitations concern the low quality of most included studies since, of the 68 included studies, 32 did not attempt to assess the consistency of results, and only 18 were considered to be at low risk of bias. Moreover, the association between the obtained phenotypes and the clinical outcomes was out of the study’s scope and should be further explored.

To our knowledge, this is the first systematic review that summarized data-driven asthma phenotypes, based on easily accessible variables, in adults. Unsupervised methods have emerged as a novel tool in adult asthma phenotyping, with the advantage of being free from a priori biases; this study provides an overview of the current state in the field, which may be useful to clinical practitioners and researchers, particularly in the understanding of the heterogeneity of asthma. The main strength of this review is the exhaustive compilation of asthma phenotypes with a detailed description of the data-driven methods used (Appendix A). Additionally, our study included an extensive literature search by applying no language or date restrictions and performing risk of bias assessment by ROBINS-I tool. The high number of included publications proves the existence of a need to classify asthma patients using data-driven methods due to the limitations of classical theory-driven approaches.

In conclusion, data-driven methods are increasingly used to derive asthma phenotypes; however, the high heterogeneity and multidimensionality found in this study suggest that both clinic and statistical expertise are required. Further research should focus on population-based samples and evaluation of longitudinal consistency of phenotypes.

## Figures and Tables

**Figure 1 diagnostics-11-00644-f001:**
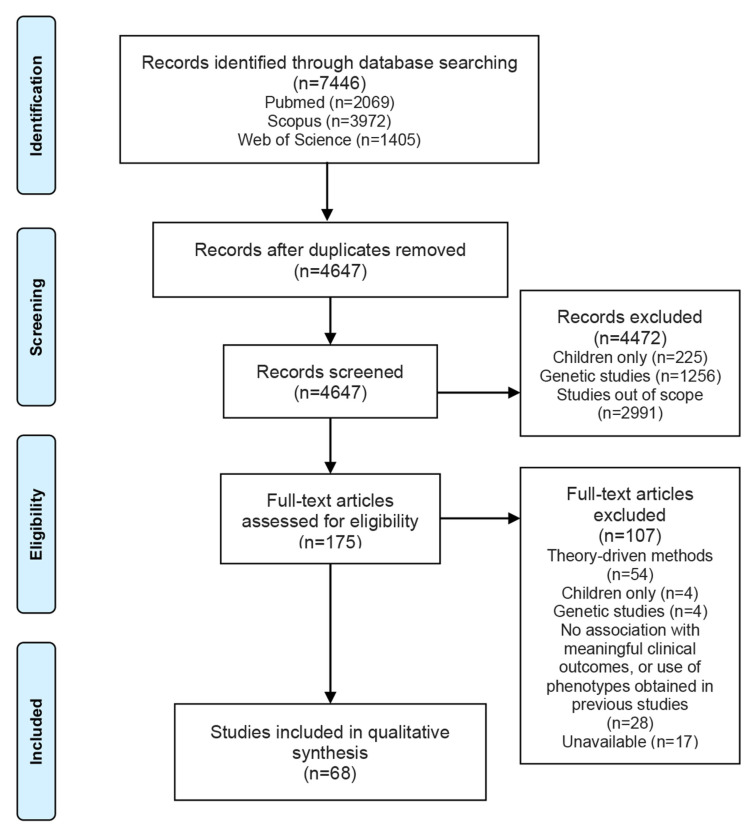
Preferred Reporting Items for Systematic Reviews and Meta-Analyses (PRISMA) flow diagram illustrating the studies’ selection process.

**Figure 2 diagnostics-11-00644-f002:**
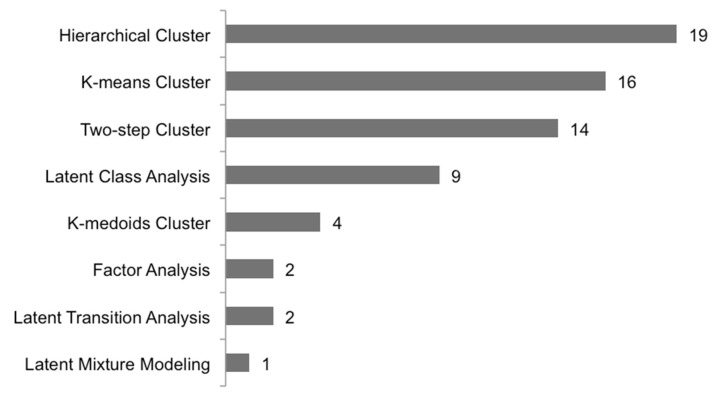
Data-driven method chosen for asthma phenotyping ordered by absolute frequency of use.

**Figure 3 diagnostics-11-00644-f003:**
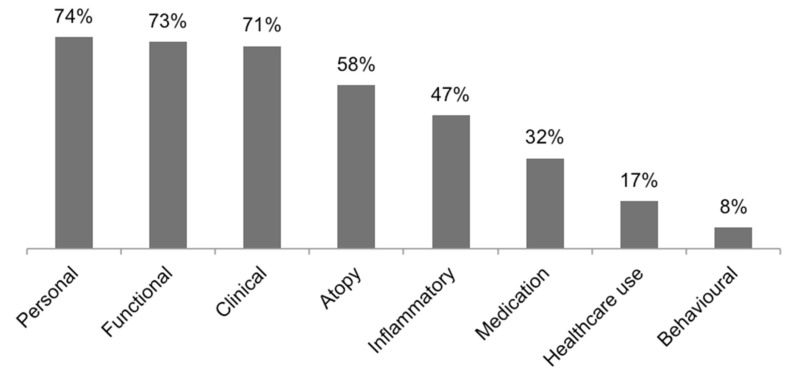
Proportion of each domain of variables in the 66 studies with retrievable chosen variables.

**Table 1 diagnostics-11-00644-t001:** List of queries used for searching online databases.

Database	Research Query
Pubmed	(phenotyp*[Title/Abstract] OR cluster*[Title/Abstract]) AND (“Asthma”[MeSH] OR asthm*[Title/Abstract]) AND (“Adult”[MeSH] OR “Adult” [Title/Abstract] OR adult*[ Title/Abstract] OR “Middle Aged”[Mesh:NoExp] OR “Aged”[Mesh:NoExp]) AND (humans[mesh:noexp] NOT animals[mesh:noexp]) NOT ((Review[ptyp] OR Meta-Analysis[ptyp] OR Letter[ptyp] OR Case Reports[ptyp]))
Scopus	(TITLE-ABS-KEY (asthm*) AND TITLE-ABS-KEY ((phenotyp* OR cluster*)) AND TITLE-ABS-KEY ((adult* OR “middle aged” OR elderly))) AND (EXCLUDE (DOCTYPE, “re”) OR EXCLUDE (DOCTYPE, “le”) OR EXCLUDE (DOCTYPE, “ed”) OR EXCLUDE (DOCTYPE, “no”) OR EXCLUDE (DOCTYPE, “ch”) OR EXCLUDE (DOCTYPE, “sh”))
Web of Science	(TS = (asthm*) AND TS = ((phenotyp* OR cluster*)) AND TS = ((adult* OR middle aged or elderly))) NOT DT = (BOOK CHAPTER OR REVIEW OR EDITORIAL MATERIAL OR NOTE OR LETTER)

**Table 2 diagnostics-11-00644-t002:** List of variables covered by each domain.

Domain	Variables
Personal	Gender, age, smoking, BMI, family history of asthma, race, education level, socioeconomic status
Functional	FEV1, FVC, FEV1/FVC, KCO or other lung function measurements, reversibility of obstruction, bronchial hyperresponsiveness
Clinical	Symptoms, exacerbations, asthma control, asthma severity scores, activity limitation, age of onset, disease duration, work-related asthma, near-fatal episode, associated comorbidities, imaging-related
Atopy	Atopic status, serum IgE, sensitization, allergen exposure, rhinitis or other allergic diseases, skin prick test, immunotherapy
Inflammatory	FeNO, blood eosinophils, and neutrophils, sputum eosinophils, and neutrophils, hsCRP
Medication	Regular medication, daily dose of prednisolone or equivalent, use of rescue bronchodilator, oral corticosteroid use
Healthcare use	Emergency department use, hospitalizations, stays in ICU, unscheduled visits to GP
Behavioral	Attitude towards the disease, perception of control, observed behavior, psychological status, confidence in doctor, stress in daily life, impact on activities in daily life

Body mass index (BMI), forced expiratory volume in 1 s (FEV1), forced vital capacity (FVC), carbon monoxide transfer coefficient (KCO), immunoglobulin E (IgE), fractional exhaled nitric oxide (FeNO), high-sensitivity C-reactive protein (hsCRP), intensive care unit (ICU), general practitioner (GP).

**Table 3 diagnostics-11-00644-t003:** Characteristics of the included studies.

Study ID (Author, Year)	Setting, Design	Inclusion Criteria in the Analysis	Number of Patients Included in the Analysis	Age	Patients’ Characteristics	Variables Used for Cluster Analysis (Number and Domains)	Method Used for Cluster Analysis
Agache, 2018 [17]	Single center (Romania), cross-sectional	Diagnosis of seasonal allergic rhinitis and asthma	57	34.12 ± 10.59	Intermittent asthma: 35 (8 were uncontrolled); Persistent asthma: 22 (10 were uncontrolled)	11 variables: personal, atopy	K-means Cluster Analysis
Alves, 2008 [18]	Single center (Brazil), cohort	Diagnosis of severe asthma, treatment-compliant	88	56 ± 12	Female: 73%; ICS in high dose: 67%; OCS: 30%; LABA: 88%	12 variables: personal, functional, clinical, atopy	Factor Analysis
Amaral, 2019 [19]	Population-based (NHANES—USA), cross-sectional	Adults (≥18 years) with current asthma	1059	N.A.	N.A.	4 variables in Model 1, 9 variables in Model 2: personal, clinical, inflammatory, health care use	Latent Class Analysis
Amaral, 2019 [20]	Population-based (ICAR—Portugal), cross-sectional	Adults (≥18 years) with and without self-reported asthma and/or rhinitis	728	43.9 ± 15.2	Female: 63% female; Non-smokers: 61%; ICS: 11%	19 variables: personal, functional, clinical, atopy, inflammatory	Latent Class Analysis
Amelink, 2013 [21]	Multicenter (Netherlands), cross-sectional	Adults (20–75 years), diagnosis of asthma after the age of 18, medication stability	200	53.9 ± 10.8	Female: 60.5%; Severe asthma: 38.5%	35 variables: personal, functional, clinical	K-means Cluster Analysis
Baptist, 2018 [22]	Multicenter (USA), cross-sectional	Age ≥ 55 years, with persistent asthma	180	65.9 ± 7.4	Male: 26.1%; Late-onset (after the age of 40): 46.7%	24 variables: personal, functional, clinical, atopy, medication	Hierarchical Cluster Analysis
Belhassen, 2016 [23]	Population-based (France), cohort	≥3 dispensations for asthma-related medication (2006–2014), aged 6–40 at third dispensation, hospitalization ≥12 months after the entry date	275	19.0 ± 11.7	Female: 47.3% female; Long-term disease status: 12.4%	3 variables: clinical (treatment)	Hierarchical Cluster Analysis
Bhargava, 2019 [15]	Single center (India), cohort	Asthma treated at primary and secondary care levels only with intermittent oral bronchodilators and steroids, and nebulization during the acute attacks, ≥6 months of follow-up, and ≥4 spirometry tests	100	33.4 ± 19.72	55% female; Asthma control according to GINA: 32% controlled, 19% partially controlled, 49% uncontrolled	N.A.	Hierarchical Cluster Analysis
Bochenek, 2014 [24]	Single center (Poland), cross-sectional	Diagnosis of aspirin-exacerbated respiratory disease	201	49.4 ± 12.4	Female: 66.6%; Intermittent asthma: 18.9%; Mild persistent asthma: 15.9%; Moderate persistent asthma: 34.8%; Severe persistent asthma: 30.3%	12 variables: personal, functional, clinical, atopy, inflammatory	Latent Class Analysis
Boudier, 2013 [25]	Population-based (ECHRS, SAPALDIA and EGEA studies), cohort	Adults, report of ever asthma	3320	35.8 ± 9.8	Female: 66.0%; Prevalence of BHR: 44.8% and 40.6% at baseline and follow-up, respectively	9 variables: functional, clinical, atopy, medication	Latent Transition Analysis//Expectation-maximization
Chanoine, 2017 [26]	Asthma-E3N study in France, nested case–control	All women who reported having ever had asthma at least once between 1992 and 2008	4328	69.6 ± 6.1	All female; Patients on maintenance therapy: 899 (13.6% with low controller-to-total asthma medication ratio)	Medication (8-year fluctuations of controller-to-total asthma medication ratio)	Latent Class Analysis
Choi, 2017 [27]	Multicenter (3 different imaging centers in the USA), cross-sectional	Diagnosis of asthma	248	NSA: 36.0 ± 12.2 SA: 46.9 ± 13.1	Nonsevere asthma: 106 (64% female); Severe asthma: 142 (63% female)	57 variables: clinical (CT imaging)	K-means Cluster Analysis
Couto, 2015 [28]	Multicenter (databases of elite athletes in Portugal and Norway), cross-sectional	Diagnosis of asthma according to criteria set by the Internal Olympic Committee to document asthma in athletes	150	25 (14–40)	Male: 71%; 91 Portuguese and 59 Norwegian	9 variables: functional, clinical, atopy, inflammatory, medication	Latent Class Analysis
Deccache, 2018 [29]	REALISE survey of adult asthma patients in 11 European countries, cross-sectional	French survey respondents	1024	34.8	Female: 66%; Active smokers: 26%; Asthma control (GINA): 17% controlled, 35% partially controlled, 48% uncontrolled	3 variables: behavioural	K-means Cluster Analysis
Delgado-Eckert, 2018 [30]	Multicenter (BIOAIR study in Europe), cohort	Diagnosis of asthma	45 (after data analysis of 138 patients)	-	Severe asthma: 76; Mild-to-moderate asthma: 62	2 variables: functional	Hierarchical Cluster Analysis
Fingleton, 2015 [31]	Cross-sectional	Symptoms of wheeze and breathlessness in the last 12 months	452	18 to 75	N.A.	13 variables: personal, functional, clinical, inflammatory	Hierarchical Cluster Analysis
Fingleton, 2017 [32]	Cross-sectional	Symptoms of wheeze and breathlessness in the last 12 months	345	55.9 ± 8.7	Male: 45.5%	12 variables: personal, functional, clinical, inflammatory	Hierarchical Cluster Analysis
Gupta, 2010 [16]	Single center (UK), cross-sectional	Severe asthma, measurable right upper lobe apical segmental bronchus, and sufficient baseline data	99	N.A.	N.A.	Unspecified (representative variables identified on factor analysis)	K-means Cluster Analysis
Haldar, 2008 [33]	Single center (UK), cross-sectional First dataset: primary-care Second dataset: secondary care, refractory asthma	Diagnosis of asthma and sufficient symptoms to warrant at least one prescription for asthma therapy in the previous 12 months	371 Primary care: 184 Secondary care: 187	Primary care: 49.2 ± 13.9 Secondary care: 43.4 ± 15.9	Female: primary care—54.4%; secondary care—65.8%	Functional, clinical, inflammatory, behavioral,	Two-step Cluster Analysis
Hsiao, 2019 [34]	Single center (Taiwan), cross-sectional	Older than 20 years, diagnosis of asthma	720	53.63 ± 17.22	Female: 58.47%	8 variables: personal, functional, atopy, inflammatory	Two-step Cluster Analysis
Ilmarinen, 2017 [35]	Single center (Finland), cohort	Diagnosis of asthma	171	N.A.	Female: 58.5%; Nonatopic: 63.5%	15 variables: personal, functional, clinical, atopy, inflammatory	Two-step Cluster Analysis
Jang, 2013 [36]	Multicenter (tertiary referral hospitals, Korea), cohort	Refractory asthma (ATS criteria)	86	39.9 ± 17.3	Female: 61.6%	5 variables: personal, functional	Two-step Cluster Analysis
Janssens, 2012 [37]	Multicenter (Belgium), Cross-sectional Two subsamples: university students, secondary care outpatient respiratory clinic	Student subsample: physician-diagnosed asthma and familiarity with asthma reliever medication; Outpatient clinic subsample: diagnosed with asthma for at least 6 months, with lung function measurement, and no other pulmonary obstructive disease	94 Student subsample: 32; Outpatient clinic subsample: 62	37.87 ± 18.56	Female: 54.26% female; Intermittent asthma: 10.64%; Mild persistent asthma: 30.85%; Moderate persistent asthma: 53.19%; Severe persistent asthma: 4.26%	6 variables: functional, clinical, medication, behavioral	Latent Transition Analysis//Expectation-maximization
Jeong, 2017 [38]	Population-based (SAPALDIA—Switzerland), cohort	Ever asthma	959	N.A.	N.A.	7 variables: personal, clinical, atopy, medication	Latent Class Analysis
Khusial, 2017 [39]	Multicenter (ACCURATE trial), randomized clinical trial	Adult asthmatics, 18–50 years old, treated in primary care, with one-year follow-up	611	39.4 ± 9.1	Female: 68.4%; Exacerbations in the past 12 months: 0.67 per patient	14 variables: personal, functional, clinical, atopy, inflammatory, medication	Hierarchical Cluster Analysis
Kim, 2018 [40]	Korean Asthma Database cohort	Non-smoking asthmatics, presence of reversible airway obstruction, airway hyperreactivity, or improvement in FEV1 >20% after 2 weeks of treatment with corticosteroids	1679 with imputed data (448 with complete data)	N.A.	N.A.	5 variables: functional (longitudinal levels of post-bronchodilator FEV1)	Two-step Cluster Analysis
Kim, 2017 [41]	Multicenter (Korea), cohort	Diagnosis of asthma, regular follow-up for over 1 year	259	56 (18–88)	Female: 81.5%	12 variables: personal, functional, atopy, infammatory	Two-step Cluster Analysis
Kim, 2013 [42]	Multicenter (Korea), two cohorts (COREA and SCH)	Asthma, ethnic Koreans, >18 years, regular follow-up and appropriate medications (GINA)	2567 COREA: 724; SCH: 4	N.A.	N.A.	6 variables: personal, functional, health care use	Two-step Cluster Analysis
Kisiel, 2020 [43]	Swedish cohort	Diagnosis of asthma	1291	54.3 ± 15.5	Female: 61.4%	14 variables: personal, clinical, atopy	K-medoids Cluster Analysis
Konno, 2015 [44]	Multicenter (Japan), cohort	Diagnosis of severe asthma (ATS criteria) for at least 1 year, ≥16 years	127	58.0 ± 13.1	Female: 59.8%; Onset age: 38.2 ± 17.7; AQLQ: 5.38 (4.79–6.21)	12 variables: personal, functional, atopy, inflammatory	Hierarchical Cluster Analysis
Konstantellou, 2015 [45]	Single center (Greece), cohort	Adult asthmatics, optimally treated for at least 6 months and adherent to therapy	170	N.A.	Persistent airflow obstruction: 35.3% (71.1% of which with criteria for severe refractory asthma vs. 4.5% in the non-persistent group)	4 variables: clinical, atopy, medication	Two-step Cluster Analysis
Labor, 2018 [46]	Single center (tertiary hospital pulmonology outpatient clinic, Croatia), cross-sectional	Physician diagnosis of asthma (GINA) at least a year before the start of the study	201	38 (26–51)	Female: 62.5%	11 variables: personal, functional, clinical, atopy	Two-step Cluster Analysis
Lee, 2017 [47]	Population-based (KNAHES and NHI claims, Korea)	Age ≥20 years and acceptable spirometry, FEV1/FVC <0.7 and FEV1 ≥60% predicted	2140	63.7 ± 11.7	Female: 29%; Under any respiratory medicine: 17.1%	6 variables: personal, functional, clinical	K-means Cluster Analysis
Lefaudeux, 2017 [48]	U-BIOPRED cohort	Diagnosis of asthma	418 (266 in training set, 152 in validation set)	N.A.	N.A:	8 variables: personal, functional, clinical, medication	K-medoids Cluster Analysis
Lemiere, 2014 [49]	Single center (tertiary center, Canada), cohort (2006–2012)	Subjects investigated for possible occupational asthma with a positive specific inhalation challenge	73	40.05 ± 10.3	Male: 61.2%	6 variables: personal, atopy, inflammatory, medication	Two-step Cluster Analysis
Loureiro, 2015 [8]	Single center (outpatient clinic, Portugal), cross-sectional	Asthmatics, age between 18 and 79 years	57	45.6 ± 18.0	Female: 73.7%; Severe exacerbation (previous year): 52.6%; Severe asthma (WHO): 57.9%	22 variables: personal, functional, clinical, atopy, inflammatory, medication	Hierarchical Cluster Analysis
Loza, 2016 [9]	ADEPT and U-BIOPRED studies, cross-sectional and cohort	Diagnosis of asthma	156	N.A.	N.A.	9 variables: functional, clinical, inflammatory	K-medoids Cluster Analysis
Makikyro, 2017 [50]	Population-based (Northern Finnish Asthma Study), cross-sectional	Adults 17–73 years old who had asthma and lived in Northern Finland, diagnosis of asthma according to the criteria of The Social Insurance Institution of Finland	1995	<30: 212 30–59: 1268 ≥60: 515	Female: 65.3%	5 variables: medication, health care use; 5 covariates: personal, clinical, atopy	Latent Class Analysis
Moore, 2010 [51]	Multicenter (USA), Severe Asthma Research Program (SARP) cohort	Nonsmoking asthmatics who met the ATS definition of severe asthma, older than 12 years of age	726	37 ± 14	Female: 66%	34 variables: personal, functional, clinical, atopy, medication, health care use	Hierarchical Cluster Analysis
Moore, 2014 [52]	Multicenter (USA), Severe Asthma Research Program (SARP) cohort	Nonsmoking asthmatics with severe or mild-to-moderate disease	423 (severe—126; not severe—297)	Severe: 41 ± 14; Not severe: 34 ± 13	Female: severe—56%; not severe—66%	15 variables: personal, functional, inflammatory, medication, health care use	Factor Analysis
Musk, 2011 [53]	Random sample from the electoral register for the district of Busselton, Western Australia, cross-sectional	Adults	1969	54 ± 17	Female: 50.6%; Reported “doctor-diagnosed asthma”: 18%; Reported wheeze: 24%; Reported “doctor-diagnosed bronchitis”: 20%; Atopic: ~50%; Never smoked: 51%	10 variables: personal, functional, atopy, inflammatory	K-means Cluster Analysis
Nagasaki, 2014 [54]	Multicenter (Japan),	Adult patients with stable asthma, receiving ICS therapy for at least 4 years and had undergone at least 3 pulmonary function tests	224	62.3 ± 13.7	Male/female: 53/171; FEV1 measurements: 16.26 ± 13.9; Follow-up period: 8.0 ± 4.5 years	7 variables: personal, functional, clinical, atopy, inflammatory	Hierarchical Cluster Analysis
Newby, 2014 [55]	Multicenter (British Thoracic Society Severe refractory Asthma Registry), cohort	Diagnosis of asthma, at least 1 year of follow-up	349	21 ± 18	Female: 63.6%	23 variables: personal, functional, clinical, atopy, inflammatory, medication, health care use	Two-step Cluster Analysis
Oh, 2020 [56]	Single center (Korea), cohort	Diagnosis of asthma	590	N.A.	N.A.	Clinical, inflammatory (routine blood test results at enrollment)	K-means Cluster Analysis
Park, 2015 [57]	Multicenter (Korea), primary cohort; Secondary cohort to assess generalizability (COREA)	Patients 65 years or older with asthma, regular medication, and controlled status (GINA)	1301 Primary Cohort: 872 Secondary Cohort: 429	75.1 ± 5.5 (in primary cohort)	Female: 52.8% (in primary cohort)	9 variables: personal, functional, clinical, atopy	K-means Cluster Analysis
Park, 2013 [58]	Multicenter (patients from the COREA cohort, Korea), cohort	Diagnosis of asthma, followed up every 3 months	724	N.A.	N.A.	6 variables: personal, functional, atopy, health care use	K-means Cluster Analysis
Park, 2019 [59]	Multicenter (patients from the COREA cohort, Korea), cohort	Diagnosis of asthma, followed up every 3 months	486	N.A.	N.A.	Functional, clinical	Latent Mixture Modeling
Qiu, 2018 [60]	Single center (Guangzhou Institute of Respiratory Disease, China), cross-sectional	Patients aged 18–65 years with respiratory symptoms that required hospitalization; Classified as severe asthma exacerbation (requirement of a course of OCS)	218	47.43 ± 13.56	Female 57.3%	21 variables: personal, functional, clinical, inflammatory	Hierarchical Cluster Analysis
Rakowski, 2019 [61]	Single center (NYU/Bellevue Hospital Asthma Clinic, USA), cohort	Adults with a primary diagnosis of asthma who had undergone a visit at the center within a 3-month period	219	59.2 ± 16	Female: 22%	Inflammatory (distribution of blood eosinophil levels)	K-means Cluster Analysis
Rootmensen, 2016 [62]	Single center (pulmonary outpatient clinic, Netherlands), cross-sectional	Over 18 years, diagnosis of asthma or COPD by pulmonary physicians, understood Dutch sufficiently to answer the questionnaires, never had consulted a pulmonary nurse	191	61 ± 15	Female: 43%; Diagnosed as having COPD: 58%; Diagnosed as having asthma: 42%	8 variables: personal, functional, atopy, inflammatory	K-means Cluster Analysis
Sakagami, 2014 [63]	Single center (outpatients of Niigata University Hospital, Japan), cohort	Diagnosis of bronchial asthma; available history of lung function and pharmacology, never-smokers	86	59.8 ± 13.2	Female/Male: 47/39	7 variables: personal, functional, atopy	Hierarchical Cluster Analysis
Schatz, 2014 [64]	TENOR: multicenter, prospective cohort (2001–2004)	Severe or difficult-to-treat asthma, ages 6 years or older	3612	N.A.	Female: 66.5%	8 variables: personal, functional, clinical, atopy	Hierarchical Cluster Analysis
Seino, 2018 [65]	Single center (outpatients of Niigata University Hospital, Japan), cross-sectional	Diagnosis of asthma, ≥16 years of age, depressive symptom-positive	128	63 (44.8–76)	Female: 65.6%	9 variables: personal, clinical, medication	Hierarchical Cluster Analysis
Sekiya, 2016 [66]	Multicenter (Japan), cross-sectional	>16 years old; hospitalization for severe or life-threatening asthma exacerbation, not complicated by pneumonia, atelectasis, or pneumothorax; SpO_2_ <90% on room air before treatment	175	57 ± 18	Female: 66%; Asthma severity: 34% intermittent, 18% mild persistent, 25% moderate persistent, 23% severe persistent	24 variables: personal, clinical, atopy, medication, health care use	K-medoids Cluster Analysis
Sendín-Hernández, 2018 [67]	Single center (Spain), cohort	Age over 14 years, asthma diagnosed following GEMA 2009, at least 1 positive skin prick test, symptoms and signs of asthma concordant with allergen exposure	225	39.56	Female: 57.3%; Mean FENO: 48.84 ppb	19 variables: personal, functional, clinical, atopy, inflammatory, medication	Hierarchical Cluster Analysis
Serrano-Pariente, 2015 [68]	Multicenter (Multicentric Life-Threatening Asthma Study—MLTAS, Spain), prospective cohort	Asthmatics ≥15 years with near-fatal asthma episode	84	51.5 ± 19.9	Female: 60%; Asthma severity (GINA): 2% intermittent, 2% mild persistent, 41% moderate persistent, 55% severe persistent	44 variables: personal, clinical, medication, health care use	Two-step Cluster Analysis
Siroux, 2011 [69]	Multicenter, cross-sectional EGEA: French case–control and family based study; ECHRS: Population-based cohort with an 8-year follow-up	Ever asthma	2446 EGEA2 sample: 1805; ECRHSII sample: 641	EGEA2 sample: 60% ≥40; ECRHSII sample: 44% ≥40	Female: EGEA2 sample—59%, ECRHSII sample—47%	14 variables: personal, functional, clinical, atopy	Latent Class Analysis
Sutherland, 2012 [70]	Multicenter (patients participating in the common run-in period of the TALC and BASALT trials), cohort	Adults (≥18 years of age) with persistent asthma, nonsmoking status	250	37.6 ± 12.5	Female: 68%	20 variables: personal, functional, clinical, inflammatory	Hierarchical Cluster Analysis
Tanaka, 2018 [71]	Multicenter (Japan), cohort	>16 years of age, requiring hospitalization due to severe or life-threatening asthma attacks with SpO_2_ < 90%; no heart failure, pneumonia, pneumothorax, or other pulmonary diseases on X-ray	190	N.A.	N.A.	Clinical	K-means Cluster Analysis
Tay, 2019 [72]	Multicenter (2 databases, Singapore), cohort	Diagnosis of asthma	420	52 ± 18	Female: 52.9%	9 variables: personal, functional, clinical, inflammatory	K-means Cluster Analysis
van der Molen, 2018 [73]	Multicenter (REALISE Europe survey), cross-sectional	Aged 18 to 50 years old, physician-confirmed asthma diagnosis, at least 2 asthma prescriptions in the last 2 years, used social media	7930	18–25: 19.2%; 26–35: 33.6%; 36–40: 17.2%; 41–50: 30.0%	Female: 61.7%; Diagnosed with asthma at least 11 years ago: 70.7%; Controlled, partially controlled, or uncontrolled asthma: 20.2%, 35.0%, and 44.8%, respectively	8 summary factors: behavioural	Latent Class Analysis
Wang, 2017 [74]	Single center (China), 12- month cohort Post hoc analysis of cohort study, which consisted of 2 parts (cross-sectional survey, prospective nonintervention cohort)	Diagnosis of asthma according to ATS and GINA criteria based on current episode symptoms, physician’s diagnosis, airway hyperresponsiveness, or at least 12% improvement in FEV1 after bronchodilator	284	39.1 ± 12.1	Female: 62%; Severe asthma (GINA): 9.9%	10 variables: personal, functional, clinical, atopy, behavioral	Two-step Cluster Analysis
Weatherall, 2009 [75]	Wellington Respiratory Survey (New Zealand), cross-sectional	Pre-bronchodilator FEV1/FVC <0.7 and/or reporting wheeze within the last 12 months	175	57.4 ± 13.5	Pre-bronchodilator FEV1/FVC <0.7 alone: 41.2%, Reported wheeze within the last 12 months: 34.4%, Met both criteria: 24.4%	9 variables: personal, functional, atopy, inflammatory	Hierarchical Cluster Analysis
Wu, 2018 [76]	Multicenter (China), prospective cohort	Nasal polyps and comorbid asthma, 16 to 68 years of age	110	47.45 ± 10.08	Female: 36.36%; Adult-onset asthma: 70.91%; Patients with NPcA had prior sinus surgery: 64.55%	12 variables: personal, clinical, atopy	Two-step Cluster Analysis
Wu, 2014 [10]	Severe Asthma Research Program, cohort	Diagnosis of asthma	378	N.A.	N.A.	112 variables clustered into 10 categories: personal, functional, clinical, atopy, inflammatory, medication, health care use	K-means Cluster Analysis
Ye, 2017 [77]	Single center (patients hospitalized by asthma exacerbation at the XinHua Hospital, China), cross-sectional	Asthma diagnosed according to GINA, aged 12–80 years	120	55 (34–63)	Female: 49.3%; Health care utilization in the last year: 8.9% hospitalized for asthma, 18.2% emergency for asthma, 42.9% outpatient, 30.0% none	21 variables: personal, functional, clinical, atopy, inflammatory, medication, health care use	Hierarchical Cluster Analysis
Youroukova, 2017 [78]	Bulgaria, cross-sectional	Moderate to severe bronchial asthma, on maintenance therapy in the last four weeks, age ≥18 years	40	46.37 ± 14.77	Female: 65%	16 variables: personal, functional, clinical, atopy, inflammatory	Hierarchical Cluster Analysis
Zaihra, 2016 [79]	Difficult asthma cohort (Montreal Chest Institute of the McGill University Health Centre, Canada)	Subjects aged 18–80 years with moderate or severe asthma (ATS criteria)	125 (48 moderate asthmatics and 77 severe asthmatics)	Moderate asthmatics: 46.6 ± 11.2; Severe asthmatics: 49.9 ± 12.6	Female: moderate asthmatics—48%, severe asthmatics—56%	Personal, functional, clinical, inflammatory	K-means Cluster Analysis

Not applicable (N.A.), inhaled corticosteroids (ICS), oral corticosteroids (OCS), long-acting β2 agonists (LABA), Global Initiative for Asthma (GINA), bronchial hyperreactivity (BHR), American Thoracic Society (ATS), forced expiratory volume in 1 s (FEV1), Asthma Quality of Life Questionnaire (AQLQ), forced vital capacity (FVC), World Health Organization (WHO), Spanish Guideline on the Management of Asthma (GEMA), chronic obstructive pulmonary disease (COPD).

**Table 4 diagnostics-11-00644-t004:** Characterization of the phenotypes obtained in each study according to the defining variables (column), with each row within each study corresponding to one phenotype.

Study ID (Author, Year)	Defining Variables of Phenotypes								
	Demographics	Comorbidities	Onset	Severity	Symptoms, Treatment	Lung Function	Atopy	Inflammation	Others
Hierarchical Cluster Analysis									
Baptist, 2018 [22]			Late						
				Mild					
							Atopic		
				Severe					
Belhassen, 2016 [23]					Less medication				
					Fixed dose inhalers				
					Free combination				
Bhargava, 2019 [15]			Childhood	Mild		Preserved	Atopic		
	Male	Overweight	Adolescent	Severe			Atopic		
	Female	Obese	Late	Severe			Least atop.		
	Female	Obese	Young age	Mild			Atopic		
Delgado-Eckert, 2018 [30]				Mild/Mod.					
				Severe					
Fingleton, 2015 [31]				Mod./Severe			Atopic		
		COPD							
		Obese							
				Mild			Atopic		
				Mild	Intermittent				
Fingleton, 2017 [32]		COPD	Late	Severe					
		COPD	Early						
							Atopic		
			Adult				Nonatopic		
			Early	Mild	Intermittent		Atopic		
Khusial, 2017 [39]			Early				Atopic		
	Female		Late						
						Reversible			
		Smokers							
					Exacerbators				
Konno, 2015 [44]			Early				Atopic	Mild eos	
		Smokers	Late			Fixed limitation		Intense Th2	
		Smokers	Late			Fixed limitation		Low Th2	
		Nonsmokers	Late					Low Th2	
	Female	Nonsmokers, high BMI	Late					Intense Th2	
Loureiro, 2015 [8]			Early	Mild			Allergic	Eosinophilic	
	Female			Moderate	Long evolution		Allergic	Mixed	
	Female, young		Early		Brittle		Allergic	No evidence	
	Female	Obese	Late	Severe	Highly sympt.			Mixed	
			Late	Severe	Long evolution	Chronic obstruction		Eosinophilic	
Moore, 2010 [51]	Female, young		Childhood			Normal	Atopic		
	Female, slightly older		Childhood				Atopic		
	Female, older								
			Childhood	Severe			Atopic		
	Female		Late				Less atopy		
Nagasaki, 2014 [54]			Late				Nonatopic	Paucigranulocytic	
			Early				Atopic		
			Late					Eosinophilic	
					Poor control	Low FEV1		Mixed granulocytic	
Qiu, 2018 [60]	Female		Early			Small degree of obstruction		Sputum neutrophilia	
	Female	Nonsmokers				Severe airflow obstruction		High sputum eosinophilia	
	Female					Moderate reduction of FEV1		Sputum neutrophilia	
	Male	Smokers				Severe airflow obstruction		High sputum eosinophilia	
Sakagami, 2014 [63]	Female						Low IgE		
	Young		Early				Atopic		
	Older		Late				Less atopic		
Schatz, 2014 [64]	Female, white		Adult				Low IgE		
							Atopy		
	Male								
	Nonwhite								
		Aspirin sensitivity							
Seino, 2018 [65]	Elderly			Severe	Poor control				Adherence barriers
	Elderly	Low BMI		Severe	Poor control				No adherence barriers
	Younger	High BMI		Not severe	Controlled				No adherence barriers
Sendín-Hernández, 2018 [67]				Mild	Intermittent		Low IgE		Without family history
				Mild			Intermediate IgE		With family history
				Mod./Severe	Needs CS and LABA		High IgE		With family history
Sutherland, 2012 [70]	Female	Nonobese							
	Male	Nonobese							
		Obese			Uncontrolled				
		Obese			Controlled				
Weatherall, 2009 [75]				Severe	Chronic bronchitis + emphysema	Variable obstruction	Atopic		
					Emphysema				
							Atopic	Eosinophilic	
						Mild obstruction			No other features
		Nonsmokers			Chronic bronchitis				
Ye, 2017 [77]			Early				Atopic		
				Moderate			Atopic		
			Late				Nonatopic		
						Fixed obstruction			
Youroukova, 2017 [78]			Late			Impaired	Nonatopic		
		Smokers	Late		High sympt., exacerbations				
		Aspirin sensitivity	Late		Symptomatic			Eosinophilic	
			Early				Atopic		
K-means Cluster Analysis									
Agache, 2010 [17]		Severe rhinitis					Polysensitization		
	Male	Severe rhinitis							Exposure to pets
							High IgE, polysensit.		
Amelink, 2013 [21]				Severe		Persistent limitation		Eosinophilic	
	Female	Obese			Symptomatic			Low sputum eos	High health care use
				Mild/Mod.	Controlled	Normal			
Choi, 2017 [27]									Normal airway, increased lung deformation
									Luminal narrowing, reduced lung deformation
									Wall thickening
									Luminal narrowing, increase in air trapping, decreased lung deformation
Deccache, 2018 [29]									Confident
									Committed
									Questing
									Concerned
Gupta, 2010 [16]				Severe	Concordant control score			Eosinophilic	Greater bronchodilator response
	Female	High BMI		Severe	High control score			Low eos	
				Severe	High control score			Low eos	
				Severe	Low control score			Eosinophilic	
Lee, 2017 [47]	Near-normal								
	Asthma								
	COPD								
		Asthmatic-overlap							
		COPD-overlap							
Musk, 2011 [53]	Male normal								
	Female normal								
	Female	Obese							
	Younger						Atopic		
	Male						Atopic	High eNO	
	Male					Poor FEV1	Atopic		
						BHR	Atopic		
Oh, 2020 [56]		High UA, T. Chol., AST, ALT, and hsCRP						High eos	
									Intermediate
		Low UA, T. Chol. and T. Bili.							
Park, 2015 [57]					Long duration	Marked obstruction			
	Female					Normal			
	Male	Smokers				Reduced			
		High BMI				Borderline			
Park, 2013 [58]		Smokers							
				Severe		Obstructive			
			Early				Atopic		
			Late	Mild					
Rakowski, 2019 [61]								Low eos	
								Intermediate eos	
								High eos	
Rootmensen, 2016 [62]	COPD without emphysema								
	COPD with emphysema								
							Allergic		
		Overlap with COPD					Atopic		
Tanaka, 2018 [71]	Young to middle-aged				Rapid exacerbation		Hypersensitive		
	Middle-aged and older				Fairly rapid exacerbation, low dyspnea				
		Smokers			Slow exacerbation, high dyspnea, chronic daily mild/mod. sympt.				
Tay, 2019 [72]	Female, Chinese		Late		Best control				
	Female, non-Chinese	Obesity			Worst control				
	Multi-ethnic						Atopic		
Wu, 2014 [10]	Healthy control subjects								
				Mild					
				Severe	Frequent, low AQLQ scores		High sensitization		
			Early			Low	Allergic	Eosinophilic	
		Nasal polyps	Late	Severe				Eosinophilic	
		Sinusitis	Early	Severe	The most symptoms	Lowest			Frequent health care use
Zaihra, 2016 [79]			Late	Severe					
	Female	High BMI		Severe					
			Early	Severe		Reduced	Atopic		
				Moderate		Good			
Two-step Cluster Analysis									
Haldar, 2008 [33]			Early				Atopic		Primary care
		Obese						Noneosinophilic	Primary care
					Benign				Primary care
			Early				Atopic		Secondary care
		Obese						Noneosinophilic	Secondary care
			Early		Symptomatic			Minimal eos	Secondary care
			Late		Few symptoms			Eosinophilic	Secondary care
Hsiao, 2019 [34]	Female	Normal BMI	Late			Normal	Nonatopic	Low neutrophils, low eos	
	Female, young adults							High eos, low neutrophils	
	Female	Obese	Late				Low IgE	High neutrophils, low eos	
	Male	Normal BMI	Late			Normal	Low IgE	Low eos	
	Male, young adults	Current smokers					Atopic	High eos	
	Male	Ex-smokers	Late					High eos	
Ilmarinen, 2017 [35]		Nonrhinitic							
		Smokers							
	Female								
		Obese							
	Adult		Early				Atopic		
Jang, 2013 [36]	Younger	Nonrhinitic				Well-preserved	Atopic	Eosinophilic	
	Younger					Severe	Low IgE	Highest total sputum cells, low eos	
	Female	Nonsmokers				High BHR		High number of sputum cells	
	Male	Smokers				Low			
Kim, 2018 [40]	Female, middle-to-old aged	High BMI		Mild					
	Female, younger			Mild			Atopic		
			Early	Mild		Mild decrease			
				Severe			Atopic	Eosinophilic	
				Severe		Persistent obstruction	Less atopic	Neutrophilic	
Kim, 2017 [41]			Early			Preserved	Atopic		
			Late			Impaired	Nonatopic		
			Early			Severely impaired	Atopic		
			Late			Well-preserved	Nonatopic		
Kim, 2013 [42]		Smokers							
				Severe		Obstructive			
			Early				Atopic		
			Late	Mild					
Konstantellou, 2015 [45]					Without high-dose ICS and OCS	Not related to persistent obstruction	Nonatopic		
					High-dose ICS and OCS	Persistent obstruction	Atopic		
					Without high-dose ICS and OCS	Not related to persistent obstruction	Atopic		
Labor, 2017 [46]							Allergic		
		Aspirin sensitivity							
			Late						
		Obese							
		Respiratory infections							
Lemiere, 2014 [49]					No subjects taking ICS	Normal	Atopic		Exposure to HMW agents
					Taking ICS	Lower	Atopic		
					Taking ICS	Lower	Less atopic		Only exposed to low molecular weight agents
Newby, 2014 [55]			Early				Atopic		
		Obese	Late						
				Least severe		Normal			
			Late					Eosinophilic	
						Obstruction			
Serrano-Pariente, 2015 [68]	Older			Severe					
					Respiratory arrest, impaired consciousness level				Mechanical ventilation
	Younger				Insufficient anti-inflammatory treatment		Sensistization to Alternaria alternate and soybean		
Wang, 2017 [74]	Male			Mild	Low exacerbation risk	Slight obstruction			
							Allergic		
	Female			Mild	Low exacerbation risk	Slight obstruction			
		Smokers				Fixed limitation			
									Low socioeconomic status
Wu, 2018 [76]		Nasal polyps					Atopic		
		Nasal polyps, Smokers							
	Older	Nasal polyps							
K-medoids Cluster Analysis									
Kisiel, 2020 [43]	Female		Early						
	Female		Adult						
	Male		Adult						
Lefaudeux, 2017 [48]				Mod./Severe	Well-controlled				
		High BMI, smokers	Late	Severe	OCS use	Obstruction			
				Severe	OCS use	Obstruction			
	Female	High BMI		Severe	Frequent exacerbations, OCS use				
Loza, 2016 [9]			Early	Mild		Normal		Low	
				Moderate		Mild reversible obstruction, BHR	Atopic	Eosinophilic	
				Mixed severity		Mild reversible obstruction		Neutrophilic	
				Severe	Uncontrolled	Severe reversible obstruction		Mixed granulocytic	
Sekiya, 2016 [66]			Younger		Severe				
	Female, elderly								
					Without baseline ICS treatment		Allergic		
	Male, elderly	COPD							
					No baseline sympt,				
Latent Class Analysis									
Amaral, 2019 [19]					Highly symptomatic	Better			
					Less symptomatic	Poor			
Amaral, 2019 [20]					Low probability of sympt.		Nonallergic		
					Nasal sympt. (very high), ocular sympt. (moderate)				
					Nasal, and ocular sympt. (high)		Allergic		
					No bronchial sympt.		Allergic		
					Nasal, bronchial, and ocular sympt. (very high) with severe nasal impairment		Nonallergic		
					Presence of bronchial sympt.		Allergic		
Bochenek, 2014 [24]				Moderate	Intensive				
				Mild	Well-controlled				Low health care use
				Severe	Poorly controlled, severe exacerbations	Obstruction			
	Female				Poorly controlled, frequent and severe exacerbations				
Chanoine, 2018 [26]					Never regularly maintenance therapy				
					Persistent high controller-to-total medication				
					Increasing controller-to-total medication				
					Initiating treatment				
					Treatment discontinuation				
Couto, 2018 [28]							Atopic		
									Sports
Jeong, 2017 [38]					Persistent, multiple sympt.				
					Symptomatic				
					Symptom-free		Atopic		
					Symptom-free		Nonatopic		
Makikyro, 2017 [50]	Female			Mild	Controlled				
	Female			Moderate	Partially controlled				
	Female			Unknown	Uncontrolled				
	Female			Severe	Uncontrolled				
	Male			Mild	Controlled				
	Male			Unknown	Uncontrolled				
	Male			Severe	Partially controlled				
Siroux, 2011 [69]			Childhood		Active, treated		Allergic		
			Adult		Active, treated				
				Mild	Inactive, untreated		Allergic		
			Adult	Mild	Inactive, untreated				
van der Molen, 2018 [73]									Confident, self-managing
									Confident, accepting
									Confident, dependent
									Concerned, confident
									Not confident
Factor Analysis									
Alves, 2008 [18]					Treatment-resistant, more nocturnal sympt. and exacerbations				
	Older				Longer duration	Persistent limitation, lower FEV1/FVC			
		Rhinosinusit is, nonsmokers				Reversible obstruction	Allergic		
		Aspirin intolerance			Near-fatal episodes				
Moore, 2014 [52]			Early	Mild/Mod.				Paucigranulocytic or eosinophilic sputum	
			Early	Mild/Mod.	OCS use			Paucigranulocytic or eosinophilic sputum	
				Mod./Severe	High doses of CS	Normal			Frequent health care use
				Mod./Severe	High doses of CS	Reduced			Frequent health care use
Latent Transition Analysis//Expectation-maximization									
Boudier, 2013 [25]					Few sympt., no treatment		Allergic		
					Few sympt., no treatment		Nonallergic		
					High sympt., treatment		Nonallergic		
					High sympt, treatment	BHR	Allergic		
					Moderate sympt.	BHR	Allergic		
					Moderate sympt.	Normal	Allergic		
					Moderate sympt., no treatment		Nonallergic		
Janssens, 2012 [37]					Well-controlled				
					Intermediate control				
					Poorly controlled				
Latent Mixture Modeling									
Park, 2019 [59]	Male, older	Smokers					Less atopic		
		Smokers					Higher IgE		
	Younger						More atopic		
	Female	Nonsmokers							

Studies are stratified by a data-driven method. Phenotypes are compiled in their full extent in Appendix A. Chronic obstructive pulmonary disease (COPD), body mass index (BMI), eosinophils (eos), forced expiratory volume in 1 s (FEV1), forced vital capacity (FVC), immunoglobulin E (IgE), corticosteroids (CS), inhaled corticosteroids (ICS), oral corticosteroids (OCS), long-acting β2 agonists (LABA), Asthma Quality of Life Questionnaire (AQLQ), exhaled nitric oxide (eNO), uric acid (UA), cholesterol (Chol.), bilirubin (Bili.), high-sensitivity C-reactive protein (hsCRP), bronchial hyperreactivity (BHR).

**Table 5 diagnostics-11-00644-t005:** Risk of bias assessment using ROBINS-I.

Study ID (Author, Year)	Confounding	Selection of Patients	Classification of Interventions	Deviations from Intended Interventions	Missing Data	Measurement of Outcomes	Selections of Reported Results	Overall
Agache, 2018 [17]	+	+	+	+	+	+	+	+
Alves, 2008 [18]	0	-	+	+	+	+	+	-
Amaral, 2019 [19]	0	+	0	0	+	+	+	0
Amaral, 2019 [20]	+	+	+	+	+	+	+	+
Amelink, 2013 [21]	0	+	+	+	+	+	+	0
Baptist, 2018 [22]	-	-	+	+	+	+	+	-
Belhassen, 2016 [23]	--	--	-	+	-	+	+	--
Bhargava, 2019 [15]	-	0	-	+	+	+	+	-
Bochenek, 2014 [24]	0	+	+	+	+	+	+	0
Boudier, 2013 [25]	+	+	+	+	+	+	+	+
Chanoine, 2017 [26]	-	+	+	+	+	+	+	-
Choi, 2017 [27]	+	+	+	+	+	+	+	+
Couto, 2015 [28]	-	+	+	+	+	+	+	-
Deccache, 2018 [29]	+	+	+	+	+	+	+	+
Delgado-Eckert, 2018 [30]	--	--	-	0	-	0	-	--
Fingleton, 2015 [31]	0	-	+	+	0	+	+	-
Fingleton, 2017 [32]	0	-	+	+	0	+	+	-
Gupta, 2010 [16]	0	0	+	+	+	+	+	0
Haldar, 2008 [33]	0	+	+	+	+	+	+	0
Hsiao, 2019 [34]	0	+	+	+	+	+	+	0
Ilmarinen, 2017 [35]	+	+	+	+	+	+	+	+
Jang, 2013 [36]	0	0	+	+	0	+	+	0
Janssens, 2012 [37]	0	+	+	+	+	+	+	0
Jeong, 2017 [38]	0	+	+	+	+	+	+	0
Khusial, 2017 [39]	+	+	+	+	+	+	+	+
Kim, 2018 [40]	0	0	+	+	0	+	+	0
Kim, 2017 [41]	-	0	+	+	+	+	+	-
Kim, 2013 [42]	-	+	+	+	+	+	+	-
Kisiel, 2020 [43]	0	+	+	+	+	+	+	0
Konno, 2015 [44]	0	0	+	+	+	+	+	0
Konstantellou, 2015 [45]	0	0	+	+	+	+	+	0
Labor, 2018 [46]	+	+	+	+	+	+	+	+
Lee, 2017 [47]	0	+	+	+	+	+	+	0
Lefaudeux, 2017 [48]	+	+	+	+	+	+	+	+
Lemiere, 2014 [49]	0	0	+	+	+	+	+	0
Loureiro, 2015 [8]	+	+	+	+	+	+	+	+
Loza, 2016 [9]	0	+	+	+	+	+	+	0
Makikyro, 2017 [50]	0	+	+	+	+	+	+	0
Moore, 2010 [51]	+	+	+	+	+	+	+	+
Moore, 2014 [52]	+	+	+	+	+	+	+	+
Musk, 2011 [53]	+	+	+	+	+	+	+	+
Nagasaki, 2014 [54]	0	+	+	+	+	+	+	0
Newby, 2014 [55]	+	+	+	+	+	+	+	+
Oh, 2020 [56]	-	0	+	+	+	+	+	-
Park, 2015 [57]	0	+	+	+	+	+	+	0
Park, 2013 [58]	0	+	+	+	0	+	+	0
Park, 2019 [59]	--	+	+	+	+	+	+	--
Qiu, 2018 [60]	-	0	+	+	+	+	+	-
Rakowski, 2019 [61]	-	+	-	+	+	+	+	-
Rootmensen, 2016 [62]	+	+	+	0	+	+	+	0
Sakagami, 2014 [63]	0	0	+	+	+	+	+	0
Schatz, 2014 [64]	0	+	+	0	+	+	+	0
Seino, 2018 [65]	0	+	+	0	+	+	+	0
Sekiya, 2016 [66]	+	+	+	+	+	+	+	+
Sendín-Hernández, 2018 [67]	+	+	+	+	+	+	+	+
Serrano-Pariente, 2015 [68]	0	+	+	+	+	+	+	0
Siroux, 2011 [69]	+	+	+	+	+	+	+	+
Sutherland, 2012 [70]	+	+	+	+	0	+	+	0
Tanaka, 2018 [71]	0	+	-	+	0	+	+	-
Tay, 2019 [72]	0	+	+	+	+	+	+	0
van der Molen, 2018 [73]	-	+	+	+	+	-	+	-
Wang, 2017 [74]	0	+	0	+	+	+	+	0
Weatherall, 2009 [75]	0	+	+	+	0	+	+	0
Wu, 2018 [76]	-	0	+	+	+	+	+	-
Wu, 2014 [10]	+	0	+	+	+	+	+	0
Ye, 2017 [77]	+	+	+	+	+	+	+	+
Youroukova, 2017 [78]	-	+	+	+	+	+	+	-
Zaihra, 2016 [79]	-	+	+	+	+	+	+	-

Caption: + = Low | 0 = Moderate | - = Serious | -- = Critical.

## Data Availability

Not applicable.

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
