# Peer review of "A Systematic Review of Asthma Phenotypes Derived by Data-Driven Methods"

_diagnostics, 2021, doi:10.3390/diagnostics11040644_

Round 1
Reviewer 1 Report
Thanks for sending me this interesting manuscript for reviewing. The authors did a systematic review to summarize how Data-Driven Methods define and diagnose clinical Asthma Phenotypes. I think the work is novel and meaningful. The structure of the manuscript is systematic, and the author provides proper background, reference, and discussion that support the review well.
Here are some comments that might facilitate the understanding of this review:
The last search was on 21 August 2020 and it was more than half of a year. I recommended the author update the search.
I recommended a vertical layout for table3 as it is too narrow for the information to be clear. Please check with the editor if this is possible.
Table4 is providing the same information from the columns of table3. This is an overlap.
table5 is difficult to understand, please add some notes to explain it.
The row of the 2nd column of TableA1 is not clearly distinguished, so it is hard to see which terms belong to which study.
Author Response
Response to Reviewer 1 Comments
We thank the reviewers for the valuable comments that contributed to improve this manuscript. In this document, we address the issues that were raised and give a point-by-point response to the comments and concerns of reviewer 1.
Point 1: Thanks for sending me this interesting manuscript for reviewing. The authors did a systematic review to summarize how Data-Driven Methods define and diagnose clinical Asthma Phenotypes. I think the work is novel and meaningful. The structure of the manuscript is systematic, and the author provides proper background, reference, and discussion that support the review well.
Here are some comments that might facilitate the understanding of this review:
The last search was on 21 August 2020 and it was more than half of a year. I recommended the author update the search.
Response 1: We thank the reviewer for the comment and the suggestion. We have now updated the research and added the information in the manuscript (lines 95-96: “first search in August 2020; updated in March 2021”). Although we now obtained 7446 studies, of which 279 were duplicates (lines 137-138), we did not find any additional study meeting eligibility criteria (i.e., the same 68 studies of data-driven asthma phenotypes studies were included) (lines 146-147). Our flowchart was updated accordingly (Figure 1).
Point 2: I recommended a vertical layout for table3 as it is too narrow for the information to be clear. Please check with the editor if this is possible.
Response 2: We thank the suggestion and we now provide a vertical layout for Table 3.
Point 3: Table4 is providing the same information from the columns of table3. This is an overlap.
Response 3: We do acknowledge the overlap. Nevertheless, we consider that Table 4 portrays the variables used by each study in a more visually accessible manner. Table 4 was moved to Appendix A.
Point 4: table5 is difficult to understand, please add some notes to explain it.
Response 4: We thank you for the comment. Table 5 (now Table 4) represented the input variables used to obtain phenotypes in each study. To clarify the reader we changed the title to: “Characterization of the phenotypes obtained in each study according to the defining variables (column), with each row within each study corresponding to one phenotype” (lines: 235-236)
Point 5: The row of the 2 column of TableA1 is not clearly distinguished, so it is hard to see which terms belong to which study.
Response 5: Thank you for the comment. We rectified the table accordingly.
Reviewer 2 Report
This is an interesting paper revising asthma phenotypes derived by data driven methods. The paper is well written, and globally informative, however I consider it too targeted for systematic review, and the clinical part was totally ignored; so I kindly ask the authors to provide some more completions and clarifications in some points and to discuss the clinical part also.
Generally asthma as a disease is not well discussed (what kind of disease is, how it manifests, symptoms, causes of occurrence, diagnostic methods, both classical and modern (based on analytical instrumentation)).
The phenotypes are not proper discussed, but only mentioned.
Please mention the novelty of the article and to who is addressed also.
I miss some discussion of comparing smokers and non-smokers with asthma, and I consider it important because smoking really is a driving force in these pathologies.
Line 202 – the word “each” appears two times
Author Response
Response to Reviewer 2 Comments
We thank the reviewers for the valuable comments that contributed to improve this manuscript. In this document, we address the issues that were raised and give a point-by-point response to the comments and concerns of reviewer 2.
Point 1: This is an interesting paper revising asthma phenotypes derived by data driven methods. The paper is well written, and globally informative, however I consider it too targeted for systematic review, and the clinical part was totally ignored; so I kindly ask the authors to provide some more completions and clarifications in some points and to discuss the clinical part also.
Generally asthma as a disease is not well discussed (what kind of disease is, how it manifests, symptoms, causes of occurrence, diagnostic methods, both classical and modern (based on analytical instrumentation)).
Response 1: We thank the reviewer for the comment suggestion, however, in this study we are focusing on data-driven asthma phenotypes rather than provide a narrative review for asthma. That is the reason we focused the introduction on the novel statistical methods that allow the researchers to unravel hidden characteristics of the disease. We consider this systematic review important in the process of providing future clinical use to asthma phenotyping studies. The high number of included studies (n=68) and the diversity in this field of research proves the importance of this paper. Nevertheless, we added the following sentence in the introduction: “Asthma is a chronic inflammatory disease of the airways characterized by at least partially reversible airway obstruction and bronchial hyper-responsiveness [1,2]” (lines 47-49)
Point 3: The phenotypes are not proper discussed, but only mentioned.
Response 3: We do acknowledge that the discussion of each of the 291 asthma phenotypes obtained in this review and the respective clinical outcomes should be addressed in future studies. However, this is beyond the scope of our study, as we aimed to summarize asthma phenotypes derived with data-driven methods in adults and characterize them according to the input variables. The discussion for this issue is stated in lines 242-303. Nevertheless, we now added in the limitation the following text: “Moreover, the association between the obtained phenotypes and the clinical outcomes was out of the study’s scope and it should be further explored.” (lines 376-378)
Point 4: Please mention the novelty of the article and to who is addressed also.
Response 4: We thank the suggestion and acknowledge that we can emphasize more what is written in the manuscript in lines 323-325 (“To our knowledge, this is the first systematic review that summarized data-driven asthma phenotypes, based on easily accessible variables, in adults”). Therefore, we added the following sentence in the Discussion section: “Unsupervised methods have emerged as a novel tool in adult asthma phenotyping, with the advantage of being free from a priori biases; this study provides an overview of the current state in the field, which may be useful to clinical practitioners and researchers, particularly in the understanding of the heterogeneity of asthma” (lines: 381-385).
Point 5: I miss some discussion of comparing smokers and non-smokers with asthma, and I consider it important because smoking really is a driving force in these pathologies.
Response 5: We acknowledge that cigarette smoking and asthma interact to induce important adverse effects on clinical, prognostic and therapeutic outcomes. However, after we identified a high number of input variables used to derive data-driven asthma phenotypes, we combine the variables into 3 major asthma-related domains of easily measurable clinical variables (personal, functional and clinical) and discuss them according to the domains, because all the three domains are considered important in the development, assessment, and management in asthma. Nevertheless, we added a paragraph discussing smoking clusters (lines 342-346).
Point 6: Line 202 – the word “each” appears two times.
Response 6: Thank you for noticing the typo, we rectified accordingly.
Round 2
Reviewer 2 Report
The authors addressed my requirements, and justified reasonable all their answers. The manuscript was improved. I suggest to be accepted in the present form.